# Environmentally Friendly and Broad–Spectrum Antibacterial Poly(hexamethylene guanidine)–Modified Polypropylene and Its Antifouling Application

**DOI:** 10.3390/polym15061521

**Published:** 2023-03-19

**Authors:** Biwei Qiu, Meng Wang, Wenwen Yu, Shouhu Li, Wenyang Zhang, Shuting Wang, Jiangao Shi

**Affiliations:** 1School of Materials and Chemistry, University of Shanghai for Science and Technology, Shanghai 200093, China; 2East China Sea Fisheries Research Institute, Chinese Academy of Fishery Sciences, Shanghai 200090, China; 3College of Materials Science and Engineering, Taiyuan University of Technology, Taiyuan 030024, China

**Keywords:** fishing net, antifouling, poly(hexamethylene guanidine), environmentally friendly

## Abstract

Biological fouling is one of the main reasons that limits the application of traditional polypropylene (PP) fishing nets in aquaculture. Here, a new environmentally friendly and broad–spectrum antibacterial agent called cationic poly(hexamethylene guanidine) (PHMG) was grafted onto PP molecular chains via permanent chemical bonding to inhibit the biological fouling. The antibacterial monofilaments were obtained by blending different contents of PP–g–PHMG with PP by melt spinning. FTIR results found PHMG to be stably present in the mixed monofilaments after high–temperature melt spinning molding. The crystallinity, relaxation behavior, mechanical properties, water absorptivity, and antibacterial and antifouling efficiencies of the PP–g–PHMG/PP blends were strongly dependent on PP–g–PHMG. The crystallinity increased with increasing PP–g–PHMG content. Adding PP–g–PHMG improved the breaking strength, knotting strength, and elongation at the break for all ratios of PP–g–PHMG/PP blends. However, the water absorption caused by PHMG is low, ranging between 2.48% and 3.45% for the PP–g–PHMG/PP monofilaments. The monofilaments showed excellent nonleaching antimicrobial activities against *Staphylococcus aureus* and *Escherichia coli*. The electrostatic adsorption of the negatively charged bacteria and the destruction of their cell membrane allowed the growth inhibition to reach 99.69% with a PP–g–PHMG content of 40%. The marine fish farming experiment also showed a long–term antifouling effect.

## 1. Introduction

Polypropylene (PP) fiber is the lightest of all synthetic fibers and is widely used in fishing materials, such as rope nets and cables, purse seines, and aquaculture cages [1,2,3]. In the aquaculture process, the surface of traditional PP materials easily adheres to marine organisms, blocking the mesh and affecting the exchange of water inside and outside the cage, which leads to the death of aquaculture fish, bringing huge economic losses to fishermen [4,5].

Existing antifouling technology for netting and cages in fish aquaculture mainly involves manual cleaning, mechanical cleaning, biological antifouling metal net clothing, and antifouling coating [6]. However, manual removal is not only laborious and time–consuming but also harmful to fishing nets. At present, advanced net clothing mechanical cleaning equipment can complete automatic cleaning, but this mechanical cleaning equipment is expensive [7,8]. The antifouling coating method is the most widely used antifouling method in fishing nets. For example, Kartal et al. [9] prepared multilayer nanocomposite films based on nanoiron, ZnO, Cu_2_O, zinc boron, and Econea on cationized ultrahigh molecular weight polyethylene (UHMWPE) fishing nets by using layer–by–layer molecular self–assembly technology. The nets covered with the Econea^®^/PDDA16 nanoparticles exhibited better antifouling properties than those covered with ZnO, Cu_2_O, and zinc borate. P. Muhamed Ashraf et al. [10] reported a microwave–assisted in situ synthesis of a nanocopper oxide–incorporated polyethylene glycol methacrylate–based hydrogel over a nylon fishing cage net, and the treated netting material exposed for 90 days to the estuarine environment exhibited excellent fouling resistance with the lowest biomass accumulation. However, problems such as antifoulant release may occur in practical situations, and some antifouling coatings are toxic, which harms marine organisms. Furthermore, antifouling coatings easily break off in harsh and complex marine environments, thus losing their antifouling effect [11,12,13,14].

Adding antimicrobial agents is also a means of antifouling. Antibacterial agents generally include natural antibacterial agents, inorganic antibacterial agents, and organic antibacterial agents [15]. Natural antimicrobials are extracted from animals and plants in nature and have the advantages of wide sourcing, safety, and nontoxicity [16,17]. For instance, Strasakova et al. [18] modified polypropylene with different concentrations of caraway essential oil and fixed it on talcum powder to obtain an antibacterial PP composite. When the concentration of essential oil was 4.9 ± 0.2 wt% and higher, it had significant antibacterial activity against *Staphylococcus aureus* and *Escherichia coli*. Belkhir et al. [19] mixed hydrolyzed casein, a natural material, into the PP matrix by melting extrusion. It was then processed by the melt spinning process to obtain multifilaments. The textiles showed strong antibacterial activity against Gram–positive and Gram–negative bacterial strains. However, the extraction process is complicated, and the heat resistance temperature is low and easy to decompose, hence limiting its practical application process.

Inorganic antibacterial agents are mainly Ag, Cu, Zn, and TiO_2_. These inorganic antibacterial agents usually have a broad–spectrum antibacterial effect. When the minimum antibacterial concentration is relatively low, the antibacterial effect is apparent. Furthermore, the high heat resistance temperature of antibacterial agents can be added to the matrix without fear of decomposition [20,21]. Zhang et al. [22] added inorganic antibacterial rGO and ZnO nanomaterials to the PES matrix and then prepared surface–grafted polybielectrolyte hydrogels (rGO/ZnO–z–PES) with low contamination and obtained a highly efficient bifunctional PES ultrafiltration membrane. Through dynamic filtration experiments, it was confirmed that the rGO/ZnO–z–PES membrane has high antifouling and antifouling performance, and it is higher than the rGO/Zno–PES membrane and the original PES membrane. However, the presence of inorganic particles in the matrix leads to compatibility issues. Furthermore, excessive addition appears as an agglomeration phenomenon and reduces the mechanical properties of materials. Inorganic particles are released rapidly in the actual use process because these inorganic particles are usually toxic and cause explosive pollution to the marine environment, affecting the survival and growth of marine organisms. Meanwhile, the price of inorganic antimicrobials has risen dramatically, which increases the cost of aquaculture when applied to fisheries [23,24,25,26].

Organic antibacterial agents are mainly divided into organic small–molecule antibacterial agents and organic polycationic antibacterial agents. Examples of organic small–molecule antibacterial agents are quaternary ammonium salts, quaternary phosphorus salts, guanidines, imidazoles, and thiophene. Organic small–molecule antimicrobials come from a wide variety of sources and are efficient and convenient to use. However, adding organic small–molecule antimicrobials into the matrix as antibacterial agents causes problems such as poor heat resistance, easy decomposition, and easy precipitation [27,28,29]. Most of the antibacterial agents mentioned above are added to the matrix by physical blending. However, these modification methods have the leaching characteristics of antibacterial agents, so they cannot achieve the purpose of permanent antibacterial and antifouling. The chemical grafting method can achieve the necessary changes in the surface properties of materials while maintaining the bulk properties of the materials [30]. The organic cationic polymers can be grafted onto the substrate by chemical bonding and durably antibacterial materials can be obtained [31,32].

In previous research work, our research team studied long–term antifouling materials and finally discovered an effective cationic antibacterial polymer called poly(hexamethylene guanidine) (PHMG). PHMG is an organic antibacterial agent with environmentally friendly and broad–spectrum antibacterial properties. The main antibacterial mechanism is the ability of PHMG to adsorb to the negatively charged membrane surface and quickly enter the plasma membrane, increasing the permeability of the membrane, leading to the formation of local pores and the exportation of intracellular substances and thus a loss of activity. In addition, PHMG may have an effect on DNA and cellular proteins [33]. Furthermore, the melting point of PHMG is high, which means that heat degradation can be avoided by the materials during high–temperature spinning [34,35].

However, individually added PHMG antibacterial agents can easily migrate and are easily soluble in water, resulting in unstable antibacterial effects. In this work, PHMG was grafted onto PP, which ensured the stability and uniform dispersion of the antimicrobial agents. Here, a series of PP–g–PHMG/PP monofilaments with different ratios was prepared by the melt spinning method to obtain the long–term antibacterial performance and stability of polypropylene monofilaments, so as to improve the antipollution performance of the materials while maintaining or even improving the mechanical properties. Then, the relationship between the structure and performance was explored.

## 2. Materials and Methods

### 2.1. Materials

The PHMG grafting rate of PP–g–PHMG was about 10%. It was purchased from Guilin Prenovo Antibacterial Materials Co., Ltd. (Guilin, China). The PP–g–PHMG was prepared via an in situ melting reaction between PP–g–MAH and PHMG through a twin–screw extruder. PP S700 was supplied by Sinopec Yangzi Petrochemical Co., Ltd. (Najing, China).

### 2.2. Preparation of PP–g–PHMG/PP Monofilaments

First, PP–g–PHMG and PP were dried in a vacuum drying oven and sealed. PP–g–PHMG and PP were then premixed and extruded by a single–screw extruder after melting. According to the weight ratio of PP–g–PHMG in PP–g–PHMG/PP blends, the blends with 0%, 10%, 20%, 30%, and 40% PP–g–PHMG were successively designated as P–0, P–10, P–20, P–30, and P–40. The PP–g–PHMG/PP compounds were melted and extruded from the spinneret with an aperture of 1.0 mm by a single–screw extruder from 200 °C to 250 °C. The aspect ratio of the single screw was 1:32, and the screw speed was 22 m/min. After the prim fibers were extruded, they were set in cooling water at a temperature of 25 °C, then in a hot water tank of 98 °C, and then stretched 8.0 times by the drafting machine. Finally, these blend monofilaments with a diameter of about 0.2 mm and a linear density of about 34.2–44.2 tex were wound into a coil on the winding machine.

### 2.3. Characterization

The chemical structures of PP–g–PHMG/PP–blended monofilaments were studied by Fourier transform infrared spectroscopy (FTIR, SPECTRUM 100, PerkinElmer Co., Ltd., Waltham, MA, USA). The samples were hot–pressed with a vulcanizer at 180 °C and then cleaned with anhydrous ethanol and scanned eight times with a resolution of 4 cm^−1^. These infrared spectra were obtained within the wavenumber range of 4000–500 cm^−1^.

The melting and crystallization behavior of monofilaments was investigated using a differential scanning calorimeter (DSC, 204F1, Netzsch Instruments, Selb, Germany). Samples were tested in a nitrogen atmosphere at a heating and cooling rate of 10 °C/min between room temperature and 240 °C. The crystallinity (*X_c_*) of blends was calculated via the total enthalpy method, according to the following Equation (1):(1)$$X_{c}=\left(\frac{\Delta H_{f}^{obs}}{\Delta H_{f}^{0}}\right)\times 100$$
where (Δ*H_f_ ^obs^*) is the observed heat of fusion values and (Δ*H_f_* ^0^) is the specific enthalpy for a 100% crystalline polymer. The (Δ*H_f_* ^0^) value of 190 J/g was used for PP [36,37].

In a tensile mode with a frequency of 1 Hz and amplitude of 30 μm, all samples were performed using the dynamical mechanical analyzer (DMA, 242C, Netzsch Instruments, Selb, Germany). The sample was first refrigerated with liquid nitrogen from room temperature to −180 °C, and held at this temperature for 10 min, and then heated to 150 °C at the set heating rate of 3 °C/min. The loss factor tan *δ* can be expressed by Equation (2):(2)$$\tan\delta =E^{\prime \prime }/E^{\prime }$$
where ($E^{\prime }$) is the energy storage modulus, ($E^{\prime \prime }$) is the loss modulus, and (tan *δ*) is the dissipation factor of viscoelastic materials.

An INSTRON–4466 electronic tensile testing machine (4466, Instron Instruments, Norwood, MA, USA) was used to study the tensile properties of 500 mm long specimens at the speed of 200 mm/min at room temperature according to SC/T 5005–2014 specifications. The samples adopted the stretching mode, and the clamps distance adopted the S–shape. The results were averaged by testing more than 10 times.

The water absorption of PP–g–PHMG/PP blended monofilaments was measured by comparing the percentage of weight change before and after immersion in water. The initial weight of the monofilaments under dry conditions was weighed, and each individual monofilament was placed in a beaker of distilled water and weighed every 12 h. The percentage of water absorbed by the PP–g–PHMG/PP–blended monofilaments was calculated using the expression in Equation (3):(3)$$W_{A}\%=[(W_{1}-W_{0})/W_{0}]100$$
where (*W_A_*) stands for the water absorption, and (*W_0_*) and (*W*_1_) present the initial weight and the final weight of monofilaments [38].

The antimicrobial properties of PP–g–PHMG/PP blended monofilaments were tested by the shaking flask method and the inhibition zone method. During the test, the sample was cultured together with bacteria to verify the antibacterial mechanism of contact sterilization. Quantitative experiments on the antimicrobial properties of PP–g–PHMG/PP blended monofilaments were studied by the shaking flask method. In total, 0.10 g of the sample after UV–sterilization was added to a test tube containing 5 mL of bacterial culture medium with a controlled bacterial concentration of 10^5^ CFU/mL and oscillated for 24 h at 37 °C at 180 rpm. Apply 100 μL of the diluted bacterial solution evenly to the L.B. solid agar plate. Put the culture dish into the bacteria incubator at 37 °C. After 24 h of culture, take out the culture dish and count the bacteria on the culture dish. To calculate the percentage of reduction of bacterial colonies was used the following Equation (4) [39]:(4)$$\mathrm{Percentage}\;\mathrm{of}\;\mathrm{antimicrobial}\;\mathrm{activity}=\frac{(A-B)*100}{A}$$
where (*A*) is the number of colonies formed in the PP monofilament and (*B*) is the number of colonies formed in the PP–g–PHMG/PP monofilament.

The Gram–negative bacteria *Escherichia coli* and the Gram–positive bacteria *Staphylococcus aureus* were chosen in the inhibition zone test. First, L.B. liquid culture solution was diluted to 10^6^ CFU/mL in a centrifuge tube. Then, 100 μL of bacterial solution was measured and evenly coated on the L.B. solid agar plate. The samples were placed flat on the surface of a Petri dish containing solidified medium with sterilizing forceps, and then the size of the inhibition zone around the monofilaments was observed after the dish was incubated in a constant–temperature incubator at 37 °C for 24 h.

In the real antifouling tests, the monofilaments were twisted and then woven into the nets. The nets were fixed to the frame and placed in the East China Sea (27°59′22.63″ N, 121°14′4.28″ E) at a depth of 1.5 m to carry out the sea net hanging test. The nets were placed between January 2022 and August 2022. The nets were weighed before and after placement and their weight changes were calculated. Each net scale was taken three times, and its average value was calculated to evaluate its antifouling performance.

## 3. Results and Discussion

### 3.1. Fourier Transform Infrared (FTIR) Analysis of PP–g–PHMG/PP Monofilaments

The FTIR spectra of the PP–g–PHMG/PP monofilaments were recorded to confirm the successful grafting of PHMG. As shown in Figure 1 and Table 1, the peaks at 1456 cm^−1^ are caused by the vibrations of the –CH_2_ group, and the peaks at 1378, 1168, and 973 cm^−1^ are attributed to the vibrations of the –CH_3_ group, which are generated by PP. Compared with pure PP monofilaments, the blended monofilaments with PP–g–PHMG have an obvious peak value at 1640 cm^−1^, which is generated by the C = N groups of PHMG [40]. The peaks generated at 3285 and 3187 cm^−1^ are also attributed to the stretching vibration of the N–H group of PHMG. The characteristic peak at 1728 cm^−1^ belongs to C = O in the anhydride group, and the new peaks at 3285, 1728, and 1640 cm^−1^ appeared in the spectrum of the PP–g–PHMG/PP monofilaments, indicating the successful reaction between the anhydride group of PP–g–MAH and the amine group of PHMG [41,42]. These peaks appeared on PP–g–PHMG/PP monofilaments, illustrating that PHMG was stably present in the mixed monofilaments after undergoing high–temperature melt spinning molding. The same conclusion was reached by Cao et al. [41].

### 3.2. Effect of PP–g–PHMG on Crystallization and Relaxation Behavior

The crystallinity of PP monofilaments has a great influence on the mechanical properties of composites. The influence of PHMG grafting on the crystallization behavior of monofilaments should be explored. Therefore, the melting and crystallization behavior of monofilaments were studied by DSC. The primary heating and cooling DSC curves of PP–g–PHMG/PP–blended monofilaments are shown in Figure 2a,b, and the dependence of crystallinity (*X_c_*) on PP–g–PHMG content is plotted in Figure 2c. The melting points (*T_m_*), Δ*H_f_ ^obs^*, the crystallization temperature (*T_c_*), and the *X_c_* of PP–g–PHMG/PP monofilaments are presented in Table 2. As shown in Figure 2a, the blended monofilaments have only one melting peak, indicating only a single PP crystalline phase in this blended system. The melting points (*T_m_*) of all PP blends are close to 170 °C. The comparison of the temperature values corresponding to the peaks of the heating curves of the five monofilaments in Figure 2a and Table 2 indicates that the addition of PP–g–PHMG has a negligible effect on the melting point of the blended monofilaments. However, their crystallization behavior is quite different during the cooling process. As observed in Figure 2b and Table 2, the *T_c_* of the PP monofilament is 119.3 °C. However, the endothermic peaks of the blended monofilaments are transferred to higher temperatures. This finding means that PP–g–PHMG promotes the crystallization of PP.

The aforementioned result was further verified by calculating the crystallinity of the blended monofilament according to the melting peak area. The calculation results are shown in Table 2, and the crystallinity curve obtained according to the values is shown in Figure 2c. The addition of PP–g–PHMG indeed increased the crystallinity of the blended monofilament. Compared with the PP monofilament, when the content of PP–g–PHMG in the blended monofilaments increased to 40%, the crystallinity increased by 5.14%, a result that is consistent with the research of Li et al. [43].

The movement of the molecular chain segments was also related to the crystallinity. The relaxation behavior of PP–g–PHMG/PP blends is shown in Figure 3. The relaxation peak of the blended monofilaments was observed from –10 °C to 0 °C. This relaxation transition is called β–relaxation and corresponds to the glass transition temperature in the amorphous region of PP in the blended monofilaments [44,45]. After the introduction of PP–g–PHMG, the $T_{\beta }$ of the PP–g–PHMG/PP–blended monofilaments moved to lower temperatures. Much easier movements of the chain segments positively affect the crystallization and mechanical properties of the blends. The molecular chain segments have more opportunities to arrange crystallization, which is consistent with the crystallinity results presented above.

### 3.3. Analysis of the Relationship between Composition, Mechanical Properties, and Water Absorption

The mechanical properties of monofilaments play an important role in fishing efforts and the service life of fishing gear [46]. The tensile stress–strain curves of PP–g–PHMG/PP monofilaments and their knotted monofilaments are shown in Figure 4. Compared with pure PP, the addition of PP–g–PHMG improved the breaking strength (Figure 4a) and knotting strength (Figure 4c) of PP–g–PHMG/PP–blended monofilaments. This finding is mainly attributed to the increase in crystallinity of blended monofilaments, which is one of the important factors affecting the mechanical properties of the materials. In addition, the elongation–at–break of PP–g–PHMG/PP monofilaments and their knotted monofilaments increases with increasing PP–g–PHMG content, as shown in Figure 4b,d. Thus, the addition of PP–g–PHMG synchronously increases the strength and tensile toughness of the monofilaments. The increased ductility here may be related to the enhanced chain segment motility for the blends, which has been validated by the DMA results in Figure 4.

Water absorption by fishing nets deteriorates their mechanical properties and affects their dimensional stability. Hydrophilic PHMG has water absorptivity, which may affect the water absorption of the blended monofilament after grafting PP with PHMG. Therefore, the water absorption rates of PP and PP–g–PHMG/PP–blended monofilaments are contrasted in Figure 5. The water absorption reached equilibrium for all monofilaments after immersion for 72 h. After soaking in water for 96 h, the weight of pure PP monofilaments did not change significantly, and the water absorption rate was approximately 0%. The water absorption of PP–g–PHMG/PP–blended monofilaments increased with increasing PP–g–PHMG. This finding can be explained by the existence of PHMG in the monofilaments as a hydrophilic chain segment. With the increase in PHMG, the content of the total hydrophilic chain in the monofilament increased, and the water absorption rate increased gradually. However, the increase was relatively small, from approximately 2.48% to 3.45%. Compared with the work of Zhang et al. [47], the water absorption rate of GNWPU films by adding only 2 wt% PHMG directly is as high as 27%. It is much higher than the water absorption rate of PP–g–PHMG/PP–blended monofilaments containing 40 wt% PP–g–PHMG in our work. Thus, the grafted PHMG on PP can avoid the dramatic increase in water absorption of the blended monofilament, hence helping to preserve the configurations of aquaculture nets.

### 3.4. Antibacterial and Antifouling Properties of Fishing Net Fibers Grafted with PHMG

The efficiency of the antimicrobial activity of PP–g–PHMG/PP monofilaments was quantitatively studied using the shaking flask method in which samples were cultured with *Escherichia coli* for a certain time. Figure 6 shows typical antimicrobial photos (10^3^ CFU/mL) of P–0, P–10, P–20, P–30, and P–40 against *Escherichia coli*. Abundant bacterial colonies appeared on the PP plate, and an antibacterial effect was not observed. However, with the increase in PP–g–PHMG addition, the number of *Escherichia coli* colonies gradually decreased, showing an excellent antibacterial effect. The inhibition rate can be calculated by colony counting. As can be seen from Figure 7, the inhibition rate of monofilaments against *Escherichia coli* exceeded 95% when the content of PHMG was 30%. When the content of PP–g–PHMG was 40%, the inhibition rate reached 99.69%, showing almost the complete growth inhibition of *Escherichia coli*. The remarkable antibacterial activity of PP–g–PHMG/PP monofilaments may be attributed to the electrostatic affinity to the colonized bacteria, which can produce a biofilm. The guanidinium group of the grafted chains of PHMG is highly positively charged, whereas the bacteria are negatively charged. PP–g–PHMG is able to produce an electrostatic force to interact with the negatively charged membrane of bacteria, thus damaging the cell membrane and achieving a significant antibacterial effect. Dead bacteria can avoid the formation of biofilms and other larger organisms [33,48]. The preparation process and antibacterial mechanism of PP–g–PHMG/PP–blended monofilaments are shown in Figure 8.

Traditional leaching antibacterial agents often work effectively in antibacterial and antifouling properties. However, their easy migration leads to a faster release rate. Thus, the antibacterial effect cannot be maintained. In this work, the covalent bond binding between the PHMG and PP molecular chains conferred nonleaching antimicrobial properties to PP–g–PHMG/PP monofilaments. The nonleaching characteristics of the blended monofilaments were verified via the inhibition zone method, particularly by conducting qualitative tests on the bacteriostatic properties of PP–g–PHMG/PP–blended monofilaments with different contents (Figure 9). No obvious bacteriostatic zone was observed for pure PP monofilaments or most blended monofilaments on the medium of *Staphylococcus aureus* and *Escherichia coli*. Only when the content of PP–g–PHMG reached 40% was there a weak inhibitory effect on the two kinds of bacteria observed in the surrounding area. This finding may be explained by the precipitation of excessive PHMG in the high content of PP–g–PHMG. The excellent antibacterial effect of PP–g–PHMG/PP–blended monofilaments was verified by the aforementioned shaking flask method. Hence, this insignificant inhibition zone may be a manifestation of nonleaching characteristics for PP–g–PHMG. Similar conclusions were found by Chen [49].

The antifouling effect of actual marine aquaculture was further verified by performing tests of different PP–g–PHMG/PP fishing netting for marine fish farming. The P–0, P–20, and P–40 monofilaments were selected as representatives to be woven into fishing twine with 4 × 3 strands and then knitted into nettings with a mesh size of 40 mm. The nettings were mounted on a frame and placed in a shallow area of the East China Sea for 8 months (from January 2022 to August 2022) for a net hanging experiment to evaluate the antifouling performance. All three fishing nets demonstrated different degrees of fouling by organisms after 235 days (Figure 10). With the increase in PP–g–PHMG content, the fouling organisms attached to the fishing nets were reduced significantly. Then, the amount of attached fouling organisms was quantitatively analyzed by cutting the meshes and calculating their average mass. Compared with the original net, the net weights of P–0, P–20, and P–40 increased by 810.80 wt%, 367.32 wt%, and 228.92 wt%, respectively, and the biological weight gain of the P–40 fishing net was reduced by 581.88 wt% compared with that of P–0. This finding indicates that the fishing nets modified by PHMG have a good antifouling effect, and the long–term antifouling effect can be maintained in the net hanging experiment for eight months.

## 4. Conclusions

In this study, PP–g–PHMG/PP–blended monofilaments with different composition ratios were prepared, and their crystallinity, relaxation behavior, mechanical properties, water absorptivity, and antibacterial and antifouling features were analyzed. FTIR analysis showed that characteristic peaks of PHMG were found in the blended monofilaments, indicating that PHMG was successfully grafted onto PP and still existed after melting spinning. The analysis of thermal properties showed that the crystallinity of PP–g–PHMG/PP–blended monofilaments increased with increasing PP–g–PHMG, resulting in the enhancement of both breaking strength and knotting strength. Dynamic mechanical analysis showed that the glass transition temperature of the blended monofilament moved to a lower temperature, which improved the tensile toughness. Furthermore, the grafted PHMG on PP can avoid the dramatic increase in water absorption of the blend monofilament. The antibacterial property test showed that the blended monofilaments had an excellent nonleaching antibacterial effect. The highest antibacterial rate against *Escherichia coli* reached 99.69% when the content of PP–g–PHMG was 40 wt%. The comparison of the antifouling effect shows that the blending–modified fishing nets have a long–term antifouling effect, which provides a feasible scheme for the application of the new antifouling materials. Future researchers can also adopt the proposed method into antibacterial textiles and biomedical materials for reducing bacterial viruses and other microorganisms.

## Figures and Tables

**Figure 1 polymers-15-01521-f001:**
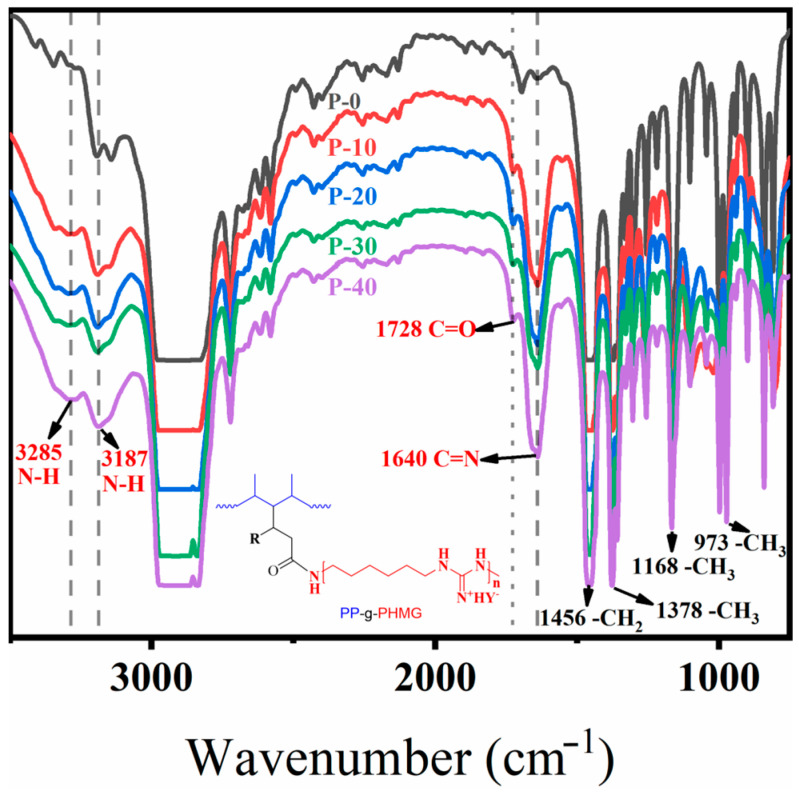
FTIR spectra of PP and PP–g–PHMG/PP–blended monofilaments.

**Figure 2 polymers-15-01521-f002:**
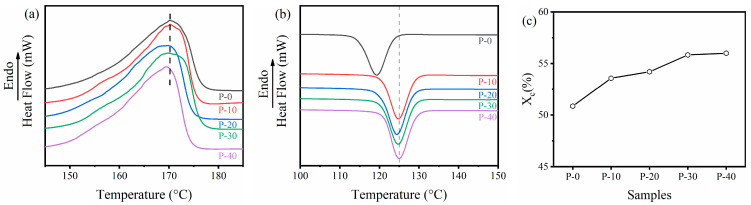
DSC curves of PP–g–PHMG/PP–blended monofilaments with different PP–g–PHMG contents under (**a**) heating and (**b**) cooling; (**c**) the dependence of crystallinity on PP–g–PHMG content.

**Figure 3 polymers-15-01521-f003:**
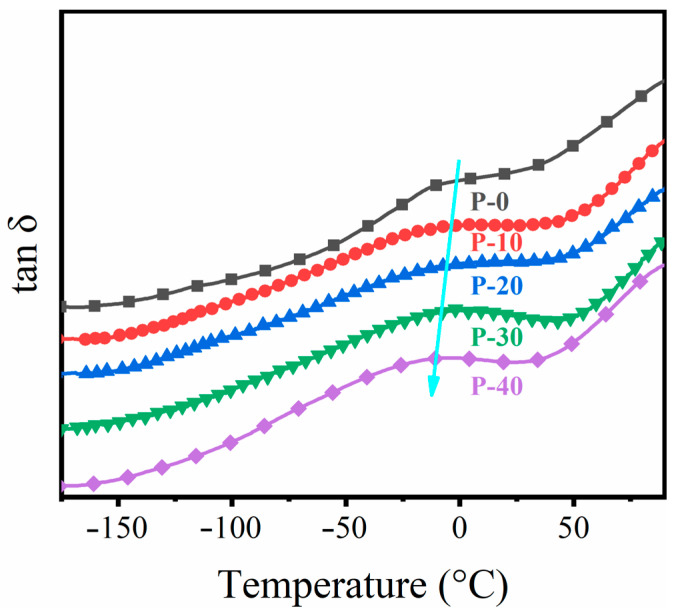
Temperature dependence of *tanδ* for PP and PP–g–PHMG/PP blended monofilaments, where arrows represented the trend of glass transition temperature.

**Figure 4 polymers-15-01521-f004:**
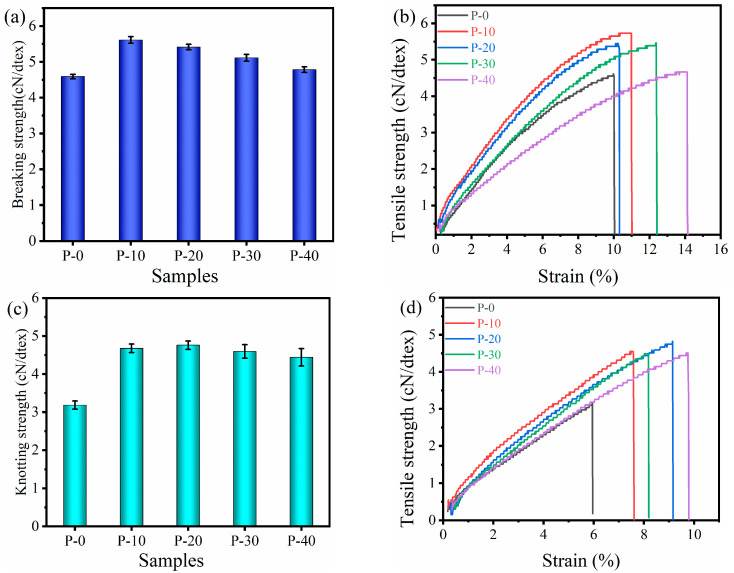
Mechanical property of the PP–g–PHMG/PP–blended monofilaments: (**a**) breaking strength, (**c**) knotting strength, and their tensile curves (**b**) and (**d**), respectively.

**Figure 5 polymers-15-01521-f005:**
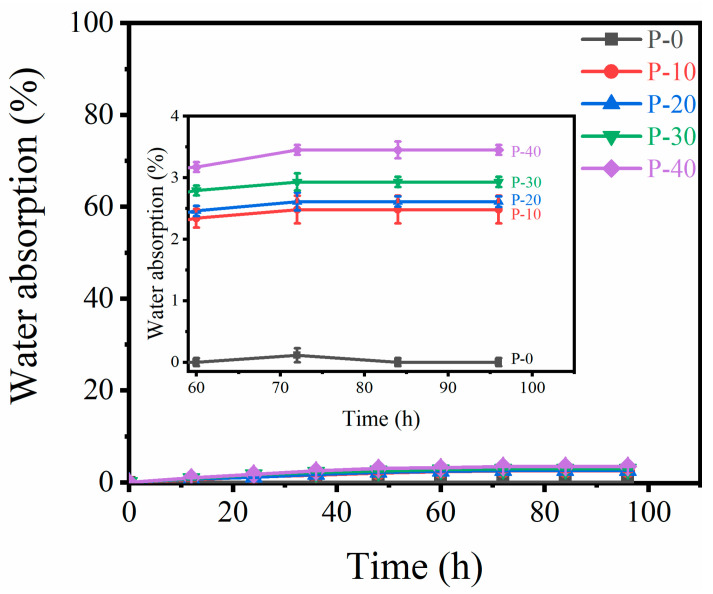
Water absorption properties of PP–g–PHMG/PP–blended monofilaments.

**Figure 6 polymers-15-01521-f006:**
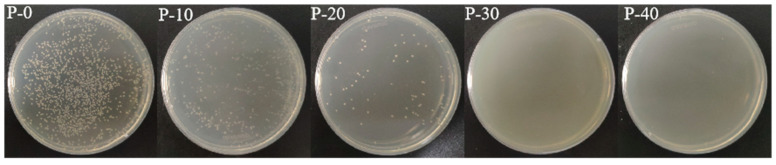
Antimicrobial activity of P–0, P–10, P–20, P–30, and P–40 monofilaments against *Escherichia coli*.

**Figure 7 polymers-15-01521-f007:**
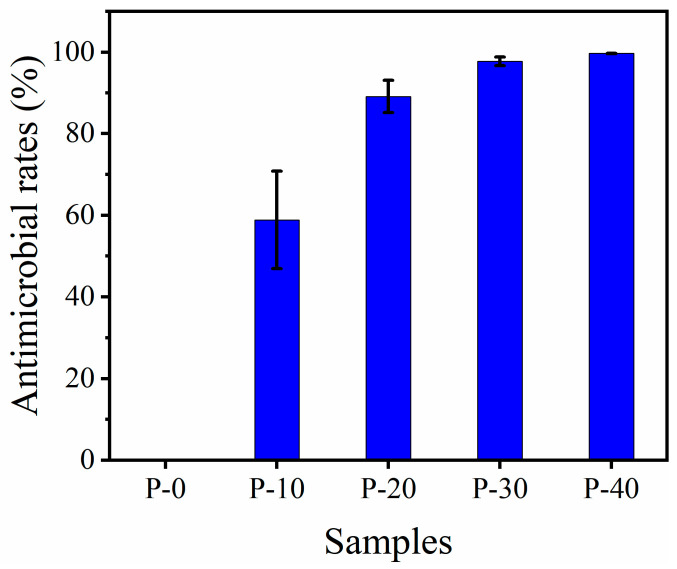
Antibacterial rate of P–0, P–10, P–20, P–30, and P–40 monofilaments against *Escherichia coli* after sample culture for 24 h.

**Figure 8 polymers-15-01521-f008:**
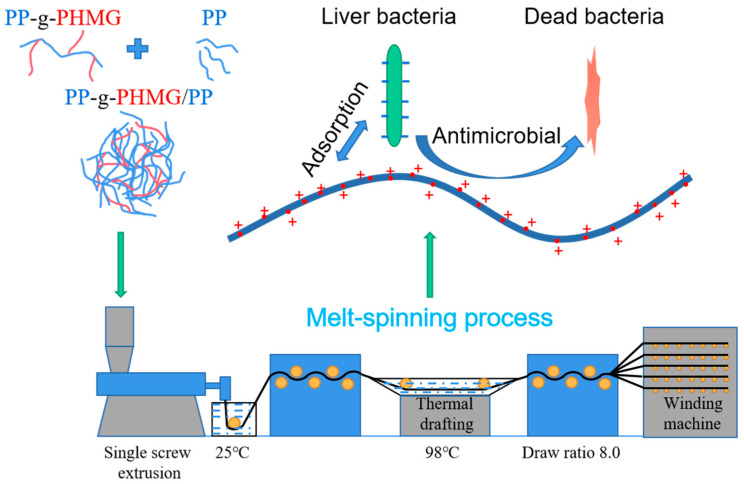
Preparation process and antibacterial mechanism of PP–g–PHMG/PP–blended monofilaments.

**Figure 9 polymers-15-01521-f009:**
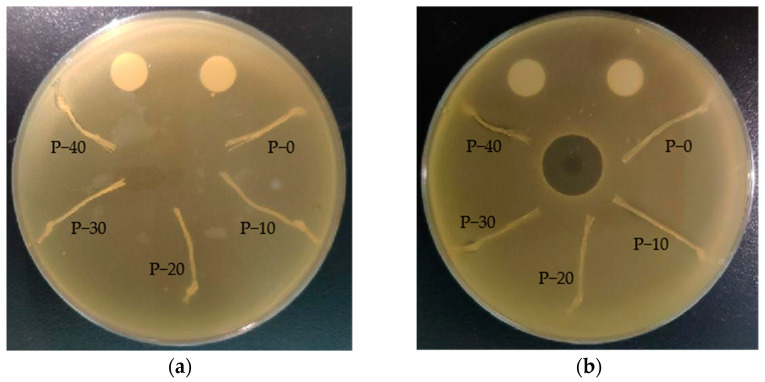
Pictures of the inhibition zone of PP–g–PHMG/PP–blended monofilaments against (**a**) *Staphylococcus aureus* and (**b**) *Escherichia coli*.

**Figure 10 polymers-15-01521-f010:**
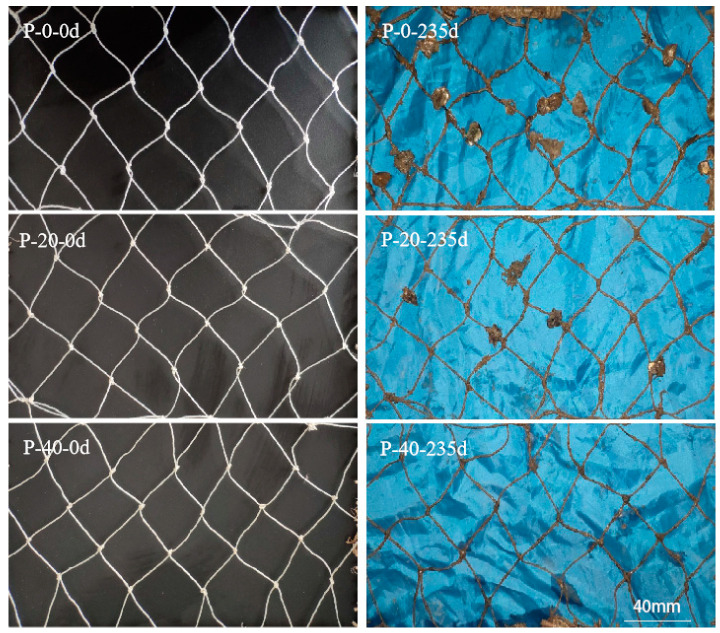
A comparison of biofouling evolution on different fishing netting materials for marine fish farming. Meshes were submerged for 235 days in the East China Sea.

**Table 1 polymers-15-01521-t001:** Wavenumbers and assigned groups of PP–g–PHMG/PP monofilaments.

Wavenumbers (cm^−1^)	Groups
3285	N–H ^1^
3187	N–H ^1^
1728	C = O ^2^
1640	C = N ^1^
1456	CH_2_ ^3^
1378	CH_3_ ^3^
1168	CH_3_ ^3^
973	CH_3_ ^3^

^1^ These groups belong to PHMG. ^2^ These groups belong to MAH. ^3^ These groups belong to PP.

**Table 2 polymers-15-01521-t002:** *T_m_*, Δ*H_f_ ^obs^*, *T_c_*, and *X_c_* of PP–g–PHMG/PP monofilaments.

Samples	*T_m_* (°C)	Δ*H_f_ ^obs^* (J/g)	*T_c_* (°C)	*X_c_* (%)
P–0	170.2	96.7	119.3	50.9
P–10	170.3	101.8	124.8	53.6
P–20	170.2	103.0	124.3	54.2
P–30	170.8	106.0	124.6	55.8
P–40	169.4	106.4	125.0	56.0

## Data Availability

Not applicable.

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
