# Peer review of "Environmentally Friendly and Broad–Spectrum Antibacterial Poly(hexamethylene guanidine)–Modified Polypropylene and Its Antifouling Application"

_polymers, 2023, doi:10.3390/polym15061521_

Round 1
Reviewer 1 Report
The paper contains interesting and valuable information. The paper can be accepted for publication after major revision. The following issues should be clarified.
The authors noted, "In this work, PHMG was grafted onto PP, which ensured the stability and uniform dispersion of the antimicrobial agents" but methodologies of the PP-g-PHMG preparation as well as characterization are absent from the paper. Additionally, it is not completely clear what the mechanism of the grafting is or what the structure of the grafted coatings is.
Information about the novelty of the paper if compared with reference 26 should be provided.
Please add a scheme where the structure of the PP-g-PHMG will be illustrated.
Please provide information from FTIR spectra in the form of a table with waves and assigned groups.
Please provide information from DSC spectra in the form of a table.
Finally, I suggest citing the paper where polypropylene was modified to obtain the valuable properties:
https://doi.org/10.1002/masy.200450638
Author Response
Response to Reviewer 1
Comments
Reviewer #1:
The paper contains interesting and valuable information. The paper can be accepted for publication after major revision. The following issues should be clarified.
Point 1: The authors noted, "In this work, PHMG was grafted onto PP, which ensured the stability and uniform dispersion of the antimicrobial agents" but methodologies of the PP-g-PHMG preparation as well as characterization are absent from the paper. Additionally, it is not completely clear what the mechanism of the grafting is or what the structure of the grafted coatings is.
Response: Thank you very much for your comments on this manuscript. In this work, the PP-g-PHMG was purchased from Gui-lin Prenovo Antibacterial Materials Co., Ltd., China. We have already made an introduction in Section 2.1 Materials. The PP-g-PHMG was prepared via an in-situ melting reaction between PP-g-MAH and PHMG according to reference 40. The structure of PP-g-PHMG has been added in Figure 1. The FTIR spectra of PP and PP-g-PHMG/PP blends were recorded to confirm the successful grafting of PHMG. As shown in Figure 1, the peaks at 3285, 3187, and 1640 cm−1 are attributed to characteristic bands of PHMG. The characteristic peak at 1728 cm−1 belongs to C=O in the anhydride group, which is the grafting group between PP and PHMG, only appearing for PP-g-PHMG/PP blends.
Point 2: Information about the novelty of the paper if compared with reference 26 should be provided.
Response: In previous work in reference 30 (original reference 26), the PE monofilament was obtained by blending PE and PP-g-PHMG. Due to poor compatibility between them, the breaking strength shows a downward trend by increasing PP-g-PHMG content, and their antibacterial efficiency was not so high. In this study, the PP-g-PHMG was added to the PP matrix. The same chain structure determines their good compatibility. It was also found that the breaking strength and knotting strength were both enhanced, compared to pure PP monofilaments. The highest antibacterial rate of PP-g-PHMG/PP blend monofilaments reached 99.69% and they showed excellent nonleaching antibacterial effect.
Point 3: Please add a scheme where the structure of the PP-g-PHMG will be illustrated.
Response: Thank you for your suggestion. The structure of PP-g-PHMG has been inserted in Figure 1.
Point 4: Please provide information from FTIR spectra in the form of a table with waves and assigned groups.
Response: Thank you for your suggestion. We have added Table 1 in Section 3.1, which indicates the peaks and assigned groups.
Point 5: Please provide information from DSC spectra in the form of a table.
Response: Thank you for your suggestion. We have added Table 2 in Section 3.2, which includes information of Tm, ΔHf obs, Tc, and Xc from DSC spectra.
Point 6: Finally, I suggest citing the paper where polypropylene was modified to obtain the valuable properties.
Response: Thank you for your suggestion. We have added two papers (reference 18 and 19) about polypropylene modification in the introduction section. The revisions have been highlighted in red.
Thanks again for your suggestions and I hope to learn more from you.
Reviewer 2 Report
Dear authors,
I congratulate you on the research done. The subject presented in the paper is original, it is relevant in the field, the conclusions are presented in accordance with the evidence and arguments in the paper. References are adequate.
I recommend you to make minor corrections.
To increase the usefulness of the article, I recommend that the introduction section be improved so as to highlight the objectives of the work.
in line 123, put c in the index ,The crystallinity (Xc)
Put a break in line 132 Where (E')is
put a break on line 135, 500mm
Author Response
Response to Reviewer 2
Comments
I congratulate you on the research done. The subject presented in the paper is original, it is relevant in the field, the conclusions are presented in accordance with the evidence and arguments in the paper. References are adequate.
I recommend you to make minor corrections.
Point 1: To increase the usefulness of the article, I recommend that the introduction section be improved so as to highlight the objectives of the work.
Response: Thank you for your suggestion. We have revised the introduction, adding citations and giving a more in-depth description of the objectives of the work. At the same time, some unnecessary descriptions were deleted. The revisions have been highlighted in red.
Point 2: in line 123, put c in the index ,The crystallinity (Xc)
Response: I am sorry for the mistakes. The crystallinity “(Xc)” has been corrected in the whole manuscript. We also checked the full text for similar silly mistakes and highlighted the changes in red.
Point 3: Put a break in line 132 Where (E')is
Response: Thank you for your suggestion. “(E')is” have been corrected to “(E') is” in the whole manuscript.
Point 4: put a break on line 135, 500mmties:
Response: I am sorry for the mistakes and thank you for your suggestion. “500mm” has been corrected to “500 mm” in the whole text.
Thanks again for your suggestions and I hope to learn more from you.
Reviewer 3 Report
In this work, the author grafted the cationic polyhexamethylene guanidine (PHGM) on polypropylene nets and evaluated its antibacterial behavior. The results and discussion seem reasonable. After careful evaluation, I think it can be accepted after a major revision.
1. The condition used to grow the bacteria is unclear (lines 151-152). Are the bacteria still growing during the antibacterial experiment??
2. To prove the polypropylene nets were grafted with PHGM successfully, SEM and XPS analysis are required.
3. Some references or characterizations should be added to prove the antibacterial mechanisms (lines 266-277).
4. Standard derivation needs to add to Figure 7. Also, all captions should be more descriptive.
5. The scale bar should be added to Figure 10. Also, the formed biofilms on polypropylene could be further analyzed (e.g., CLSM analysis). Some references can be added to strengthen the surface antibacterial and biofouling issues. For example, Journal of Colloid and Interface Science, 613 (2022): 426-434 and Journal of Membrane Science, 565 (2018): 293-302.
Author Response
Response to Reviewer 3
Comments
In this work, the author grafted the cationic polyhexamethylene guanidine (PHGM) on polypropylene nets and evaluated its antibacterial behavior. The results and discussion seem reasonable. After careful evaluation, I think it can be accepted after a major revision.
Point 1: The condition used to grow the bacteria is unclear (lines 151-152). Are the bacteria still growing during the antibacterial experiment??
Response: Thank you for your suggestion. The antibacterial experiment is an accelerated test to evaluate long-term antibacterial properties. During the test, the samples and bacteria were cultured together to verify contact sterilization with the same initial concentration. As the experiment proceeds, bacterial propagation and antimicrobial agents compete with each other. With the increased content of PP-g-PHMG, the antibacterial effect improved significantly (as shown in Figure 6 and Figure 7).
Point 2: To prove the polypropylene nets were grafted with PHGM successfully, SEM and XPS analysis are required.
Response: Thank you for your suggestion. The FTIR spectra of PP and PP-g-PHMG/PP blends were recorded to confirm the successful grafting of PHMG. As shown in Figure 1, the peaks at 3285, 3187, and 1640 cm−1 are attributed to characteristic bands of PHMG. Especially, the characteristic peak at 1728 cm−1 belongs to C=O in the anhydride group, which is the grafting group between PP and PHMG, only appearing for PP-g-PHMG/PP blends. We have supplemented the relevant description of the C=O in the revised manuscript in red.
Point 3: Some references or characterizations should be added to prove the antibacterial mechanisms (lines 266-277).
Response: Thank you for your suggestion. We have elaborated on the antibacterial mechanism of PHMG in more detail in the introduction section and added references to prove its antibacterial mechanism in section 3.4.
Point 4: Standard derivation needs to add to Figure 7. Also, all captions should be more descriptive.
Response: Thank you for your suggestion. The standard derivation has been added to Figure 7. Captions have been changed descriptively as you requested.
Point 5: The scale bar should be added to Figure 10. Also, the formed biofilms on polypropylene could be further analyzed (e.g., CLSM analysis). Some references can be added to strengthen the surface antibacterial and biofouling issues. For example, Journal of Colloid and Interface Science, 613 (2022): 426-434 and Journal of Membrane Science, 565 (2018): 293-302.
Response: Thank you for your suggestion. The scale bar has been added to Figure 10. We will analyze the formed biofilms on polypropylene using CLSM for further research. We cite your recommended papers as examples of inorganic antimicrobial agents, marked [20] and [22] in the third paragraph of the introduction.
Thanks again for your suggestions and I hope to learn more from you.
Round 2
Reviewer 1 Report
After revision, the quality of the paper was essentially improved. I have only two small remarks.
First, the reaction between the anhydride group of PP-g-MAH and the amine group of PHMG leads to the fabrication of amide bonds -C(O)-NH-. The structure of PP-g-PHMG inserted in Figure 1 should be corrected. Now is -C(O)O-NH- that is wrong, must be -C(O)-NH-.
Second, please add one more relevant reference that is strongly suitable for the paper: https://doi.org/10.1002/masy.200450638
Author Response
Response to Reviewer 1
Comments
Reviewer #1:
After revision, the quality of the paper was essentially improved. I have only two small remarks.
Point 1: First, the reaction between the anhydride group of PP-g-MAH and the amine group of PHMG leads to the fabrication of amide bonds -C(O)-NH-. The structure of PP-g-PHMG inserted in Figure 1 should be corrected. Now is -C(O)O-NH- that is wrong, must be -C(O)-NH-.
Response: Thank you very much for your comments on this manuscript. We have corrected the structure of PP-g-PHMG inserted in Figure 1 according to your requirements.
Point 2: Second, please add one more relevant reference that is strongly suitable for the paper: https://doi.org/10.1002/masy.200450638
Response: Thank you for your suggestion. We cite your recommended paper as an example of graft modification of polypropylene, marked [30] in the fifth paragraph of the introduction. The revisions have been highlighted in red.
Thanks again for your suggestions and I hope to learn more from you.
Reviewer 3 Report
Good job!
Author Response
Response to Reviewer 3
Comments
Good job!
Response: Thank you for your support for the revision of our paper, and I hope to learn more from you.